# Synthesis of Supported Heterogeneous Catalysts by Laser Ablation of Metallic Palladium in Supercritical Carbon Dioxide Medium

**DOI:** 10.3390/molecules25245807

**Published:** 2020-12-09

**Authors:** Oleg Parenago, Alexey Rybaltovsky, Evgeniy Epifanov, Andrey Shubnyi, Galina Bragina, Alexey Lazhko, Dmitry Khmelenin, Vladimir Yusupov, Nikita Minaev

**Affiliations:** 1Institute of Petrochemical Synthesis, Russian Academy of Sciences, 29 Leninsky Prospekt, 119991 Moscow, Russia; parenago@ips.ac.ru; 2Skobel’tsyn Science Research Institute of Nuclear Physics, Lomonosov Moscow State University, Leninskie Gory 1, 119991 Moscow, Russia; alex19422008@rambler.ru; 3Institute of Photon Technologies, Federal Scientific Research Centre “Crystallography and Photonics” RAS, Pionerskaya Str. 2, Troitsk, 108840 Moscow, Russia; rammic0192@gmail.com (E.E.); deerhunter9136@gmail.com (A.S.); iouss@yandex.ru (V.Y.); 4Institute of Organic Chemistry, Russian Academy of Sciences, 47 Leninsky Prospekt, 119991 Moscow, Russia; bragina@ioc.ac.ru; 5Institute of General and Inorganic Chemistry, Russian Academy of Sciences, 31 Leninsky Prospekt, 119991 Moscow, Russia; alexeylazhko@mail.ru; 6FSRC “Crystallography and Photonics” RAS, 59, Leninskiy Prospekt, 119333 Moscow, Russia; xorrunn@gmail.com

**Keywords:** supercritical carbon dioxide, laser ablation, supercritical fluid (SCF) colloids, Pd nanoparticles, Pd catalysts, hydrogenation

## Abstract

To obtain a supported heterogeneous catalyst, laser ablation of metallic palladium in supercritical carbon dioxide was performed in the presence of a carrier, microparticles of γ-alumina. The influence of the ablation process conditions—including supercritical fluid density, ablation, mixing time of the mixture, and laser wavelength—on the completeness and efficiency of the deposition of palladium particles on the surface of the carrier was studied. The obtained composites were investigated by scanning and transmission electron microscopy using energy dispersive spectroscopy. We found that palladium particles were nanosized and had a narrow size distribution (2–8 nm). The synthesized composites revealed high activity as catalysts in the liquid-phase hydrogenation of diphenylacetylene.

## 1. Introduction

In recent times, supported heterogeneous catalysts that are highly active in many organic reactions have been successfully prepared using supercritical carbon dioxide (scCO_2_) as a solvent [1]. The synthesis of such catalytic systems, described in detail in a review by Erkey [2], in the general case, involves the dissolution of a metal-containing compound in scCO_2_ and the deposition of this compound on the surface of a solid carrier of an inorganic or polymer nature. Then, the deposited compound is reduced to a metal either thermally, by molecular hydrogen, or by other reducing agents. As such, nanosized particles of various metals have been obtained by deposition on the surface of inorganic carriers, for example, on γ-alumina [3,4], mesoporous silicon, aluminum oxides [5,6,7], soot [8], graphene [9,10], carbon nanotubes [11,12], and aerogels [13,14].

For carriers of a polymer nature, this method has been used much less frequently. In the work of Watkins et al., poly-4-methylpentene-1 or polytetrafluoroethylene was used as the carrier, and the dimethyl (cyclooctadiene-1,5) platinum complex, Pt(cod)Me_2_, served as a precursor [15]. In [16], palladium, rhodium, and platinum compounds were deposited to hollow nanospherical polystyrene; in [17], rhodium and palladium compounds were deposited on crosslinked dendrimers (polypropyleneimines and polyamidoamines). The obtained metal–carrier composites were active in the catalysis of various reactions: hydrogenation of unsaturated hydrocarbons [6,10,17], allyl alcohol [16], nitrophenol [5], methanol electrooxidation [9], Suzuki–Miyaura combination reaction [4,11], water shift reaction [3], etc.

Despite the active development of research in this field, the catalysts synthesized by this method, in some cases, have a significant drawback. The metal particles obtained by the reduction of metal-containing compounds may contain impurities of anions or ligands from the initial precursors, which thereby diminishes the catalytic activity of the systems [18].

As such, the method of laser ablation of solid targets in gas or liquid media shows promise for the synthesis of nanomaterials. The general principles and main areas of nanomaterial application obtained by this method are described in two monographs in 2013 and 2016 [19,20]. Laser ablation has been widely used in the preparation of active nanoscale catalysts. Ma et al. [21] used pulsed laser ablation in the synthesis of CeO_2_/Pd composites, in which CeO_2_ plates were 20 nm and Pd particles were 9 nm. The nanocomposites were found to be highly active in the hydrogenation of 4-nitrophenol to 4-aminophenol. Using the same method, Pd nanoparticles were obtained on graphitized carbon that catalyzed the hydrogenation of nitrobenzene to aniline [22], and bimetallic Pd/Cu nanocatalyst active in the same reaction was synthesized [23]. Zhang et al. reviewed the advantages of colloidal nanoparticles obtained by laser ablation, summarizing the latest advances in the synthesis of such systems and their use in catalysis [18].

The synthesis of nanosized metal catalysts by laser ablation in supercritical media [24] is of particular interest. The main advantage of laser ablation in this medium is the ability to vary the particle sizes and their composition, and to change the pressure and/or temperature of the medium as well as the irradiation conditions (wavelength, pulse frequency, etc.). Laser ablation in the scCO_2_ medium provides various opportunities for obtaining nanocolloids and nanocomposites of metals and other elements with unique properties for use in electronics, medicine, photonics, and others. In one of their early works using this method, Saitow [25] obtained silicon nanoclusters with various electron structures. Laser ablation of highly-oriented pyrolytic graphite in the scCO_2_ medium resulted in the formation of diamondlike structures (diamondoids) [26]. Machmudah et al. [27] showed that the morphology and sizes of gold and silver nanoparticles obtained by laser ablation in scCO_2_ and deposited on a silicon plate strongly depend on the properties of the CO_2_ medium (pressure and temperature). It was found in [28] that an increase in the colloidal density of supercritical fluid from 0.24 to 0.82 g/cm^3^ caused a strong decrease in the rate of formation of silver particles, aggregation of small particles, and their faster sedimentation. The formation of composites from gold and silver nanoparticles with a polytetrafluoroethylene–vinylidene fluoride copolymer obtained by laser ablation in the scCO_2_ medium was described [29]. Singh et al. [30] used the same method for the synthesis of composite titanium oxide nanoparticles (anatase-TiO_2_) on coal; as particle size decreased, the size distribution narrowed with increasing CO_2_ pressure and temperature.

Laser ablation of metal targets in the scCO_2_ medium have shown promise in the preparation of heterogeneous catalysts, in particular, when deposited on the surface of various carriers [31,32]. The unique properties of the scCO_2_ medium, which has a very-low viscosity and high molecular mobility, provides effective drift of metal nanoparticles arising during ablation to the surface of carriers and deposition on the surface and in their pores. This method of synthesizing nanomaterials provides a number of other advantages: first, the achievement of a higher degree of ecological purity of the experiment, which excludes contamination of the environment and the synthesized material with residues of organic palladium precursors, as can occur in the case of thermochemical deposition of the obtained np Pd [2] onto the surface of the carrier; second, in the case of using scCO_2_ as a transport medium, the fixation of the synthesized nanoparticles on the features of the surface relief of the carrier microparticles is more efficient; third, a one-stage method for the synthesis of composite materials has been proposed, in which the preparation of nanoparticles and their fixation on the carrier are performed in one technological cycle, which is rarely implemented in practice.

Here, the results are reported for the laser ablation of a palladium target in a supercritical carbon dioxide medium, deposition of palladium particles on the surface of γ-alumina microparticles, and the characteristic of the obtained nanocomposite material. Data are provided on the hydrogenation of diphenylacetylene under the action of the obtained composite, characterizing its catalytic activity.

## 2. Results and Discussion

A 1-g sample of Al_2_O_3_ microparticle powder and the bar from the magnetic stirrer were placed at the bottom of the high-pressure reactor. In the middle of the reactor volume, the Pd target was installed. After sealing the reactor, CO_2_ was introduced into it to a predetermined pressure, and the reactor was heated to the required temperature. Then, using a digital camera and a window in the upper part of reactor, the optical system was adjusted so that the laser spot was located at the center of the Pd target. Then, the ablation process was started, which was recorded by video shot through a window located in the reactor top. Figure 1 shows images of the palladium target at different periods of laser exposure.

At the beginning of the process, laser radiation, practically not scattering in the reactor volume, freely reached the target surface and caused ablation (Figure 1a). After the formation of a sufficiently high concentration of Pd particles, the track of the laser beam was well traced due to its scattering on a colloidal solution of palladium particles in scCO_2_ (Figure 1b). As a result of turning on the magnetic stirrer in the beam track, the appearance of Al_2_O_3_ microparticles was observed, which caused plasma flashes when they interacted with a colloidal solution of palladium particles in the laser beam (Figure 2c). The appearance of a large concentration of carrier microparticles in the reactor volume led to strong scattering of the laser beam, which was observed through a decrease in the glow intensity near the target. Initially, the effect of the density of the medium and wavelength of laser radiation on the content of palladium on the carrier was studied (Table 1, Figure 2 and Figure 3).

A preliminary estimation of the amount of palladium removed into the scCO_2_ medium as a result of laser ablation of the target showed that ~0.1 mm^3^ of metal was released into the reactor volume during 10 min of operation (laser dose 4 × 10^6^ J/cm^2^).

During laser ablation with simultaneous stirring of the carrier, its disintegration was observed with the formation of Al_2_O_3_ particles of a smaller size compared to the initial size (250–500 μm; Figure 2). We found that the amount of destroyed particles increased with increasing time of operation of the magnetic stirrer and laser exposure, i.e., disintegration of the carrier occurred for two reasons: (1) grinding of the powders by the bar of the magnetic stirrer and (2) destruction of particles in the focus area of the laser radiation. The mechanical grinding of Al_2_O_3_ particles by the stirrer bar depends on both the amount of carrier introduced into the reactor and the density of the fluid. Thus, we observed that when Al_2_O_3_ was loaded above a certain threshold value (~1.5–2.0 g), it was significantly destroyed by the magnetic stirrer. As the density of the fluid increased, the effect of particle disintegration associated with mechanical grinding decreased. Apparently, with an increase in the density of the medium, the ability of carrier particles to be in suspension increases, decreasing their fraction at the bottom of the reactor in the region of rotation of the stirring bar.

As shown in Table 1 and Figure 3, with an increase in the density of the medium from 0.2 to 0.8 g/cm^3^, the total palladium content on the carrier decreased, while its portion in the fraction with the initial particle size increased. These results may be due to several aspects.

First, as known by the example of laser ablation of silver [28], conducting the process at a low scCO_2_ density (0.2 g/cm^3^) leads to the formation of small Ag particles (several nm). Conversely, a higher fluid density (0.8 g/cm^3^) fosters the formation of large particles and their agglomerates, which, due to their tendency to sedimentation, settles to the bottom and walls of the reactor. Thus, the overall decrease in the fraction of deposited palladium with increasing fluid density is associated with a decrease in the content of palladium nanoparticles (PdNP) in the region of action of the laser radiation and the suspended carrier particles located there.

The density of the scCO_2_ medium affects the destruction of alumina particles in the laser focusing region. It can be assumed that under low-density conditions, i.e., with a more effective laser action [28,29], the process of disintegration of carrier particles proceeds more actively, which leads to a decrease in the number of large particles in the system. As the density of the fluid increases, the destruction of Al_2_O_3_ significantly slows and the fraction of large particles (>250 μm) of the support remains predominant. As a result, these large particles with a fluid density of 0.8 g/cm^3^ provide the main matrix for the deposition of PdNP (Figure 3).

When using laser radiation with λ = 532 nm (the second harmonic of laser radiation), the time of radiation and mixing were deliberately increased (Table 1, experiment 4) to equalize the radiation dose during the first three experiments, since the radiation power for λ = 532 nm is much lower. In this case, under conditions of a high fluid density (0.8 g/cm^3^), the palladium content in the entire carrier sample (0.35 wt %) was almost two times higher compared with the experiment with λ = 1064 nm (0.2 wt %). This result is most likely associated with an increase in the radiation dose in the “free” ablation mode without scattering on Al_2_O_3_ microparticles; for sample 3, this time was 10 min; for sample 4, this time was 15 min. However, the size of the large powder fraction after treatment with laser radiation of λ = 532 nm significantly decreased. As a result, this fraction completely lacked large particles with a size >0.25 mm (Table 1). Therefore, this fraction was not used in further studies of catalytic activity. We believe that the significantly stronger grinding of fraction 4 compared to fractions 1–3 is due to two reasons. The first reason is associated with mechanical grinding: To prepare this fraction, the alumina particles were stirred with a magnetic stirrer for a longer time (15 min instead of 10 min). The second reason is due to laser action: When obtaining fraction 4, particles were acted upon by laser pulses with λ = 532 nm instead of = 1064 nm (Table 1). Shorter wavelength radiation due to the higher absorption coefficient in the alumina material contributes to greater heating and destruction of microparticles [33].

To characterize PdNP alumina deposited on microparticles by electron microscopy and tested in the hydrogenation reaction, sample 5 was prepared by laser ablation of palladium at a moderately low fluid density (0.3 g/cm3, 70 °C, and 120 bar CO_2_) for 10 min of ablation and 10 min in mixed mode. This sample was divided into fractions (0.25–0.5 mm, sample 5-1 and 0.1–0.25 mm, sample 5-2) according to the size of the particles in which the mass content of palladium was determined (Table 2).

As follows from these data and as a result of ablation during grinding the initial alumina, two fractions mainly formed: One with the initial particle size of the carrier and the second with the size of 0.1–0.25 mm. Even smaller particles (less than 0.1 mm), like dust, are not of interest as carriers for catalysts, so they were not considered in this study.

Figure 4 presents the SEM images of fractions 1 and 2 (Table 2) of sample 5. With an increase of 2 μm, against the background of aluminum oxide particles, palladium microparticles were observed in the form of bright white dots, and their number in the second fraction was visually larger than in the first fraction. The elemental analysis of particles by EDX established that the nanoparticles on the surface of the carrier were palladium. Notably, the palladium content determined by the AAS method on fraction 2 with a size of 0.1–0.25 mm was almost five times higher than the particles of the initial size (Table 2).

Then, particles of sample 5 were studied by TEM (Figure 5). Figure 5 shows that in both fractions, the palladium particles were very small (2–8 nm), whereas the particle size distribution was rather narrow. For the smaller fraction (fraction 2), the distribution of Pd particles was almost unimodal with a maximum of 3 nm on the distribution curve.

Both fractions of sample 5 were tested as catalysts. Figure 6 shows the kinetics of the reaction of liquid-phase hydrogenation of a model substrate, diphenylacetylene (DPA).

As follows from these data, palladium deposited on alumina is highly active in the hydrogenation of unsaturated hydrocarbon and, in both cases, the hydrogenation rate in the first stage, i.e., with the addition of hydrogen atom to the first unsaturated bond of the acetylene derivative molecule, is significantly higher compared to the subsequent stage. The high activity of Pd/Al_2_O_3_ composites is most likely associated with the size parameters of palladium particles on the support, which, according to TEM data, are 2–8 nm with an almost unimodal particle size distribution. The catalytic activity of the systems was determined using the slope of the kinetic curves—W_observ_, mol/min (the ratio of the number of moles of the reacted substrate to time) at both stages and is expressed as TOF, s^−1^ (Table 3).

As shown by the obtained data, despite the almost five-fold excess palladium content in the fraction of the sample of smaller size, its specific catalytic activity was almost two times lower compared to the larger fraction. As the nanoscale characteristics of palladium nanoparticles in both samples were close, this result may be connected with a smaller number of active centers of the hydrogenation process in fraction 2 due to the association of palladium particles under their sufficiently high content in the carrier matrix.

Palladium catalysts supported on carriers of various nature were used for the hydrogenation of diphenylacetylene in a number of works [34,35,36]. For this purpose, Markov et al. prepared bimetallic Pd–Cu catalysts by depositing solutions of palladium and copper salts on metal oxides, including Al_2_O_3_, followed by drying, calcining, and activating the samples in a stream of hydrogen [34,35]. The specific catalytic activity of these systems in DPA hydrogenation was 0.16–5.77 s^−1^, depending on the Cu/Pd ratio. In [36], palladium was deposited in a matrix of hyper crosslinked polystyrene, where, at the first stage in the medium of supercritical carbon dioxide, palladium compounds were deposited on the support, which were reduced to metal with molecular hydrogen in the second stage. In this case, the specific activity in DPA hydrogenation was 0.57 s^−1^. The palladium composites obtained in this work are comparable or exceed the known catalysts in activity, but the process of their preparation includes only one stage, is distinguished by the absence of solvents, and is therefore more environmentally friendly. Additionally, it allows properties of the resulting systems to be influenced by varying the conditions of laser ablation. It should be noted that the rather high activity of the proposed catalytic systems can be associated with the method of deposition of palladium particles on the surface or in the internal pores of the support from a colloidal solution in supercritical carbon dioxide. Most likely, it is this method that provides a uniform distribution of metal particles in the support matrix, which leads to high activity of the composites.

The results presented in the article are the first example of obtaining supported heterogeneous catalysts for the hydrogenation reaction by laser ablation of a metal in a supercritical carbon dioxide medium with the deposition of the resulting metal particles on the surface of the carrier.

## 3. Materials and Methods

### 3.1. Installation

A scheme of the main elements of the installation is shown in Figure 7. The experiments were conducted in a high-pressure reactor (1), into the volume of which radiation was emitted from a pulsed laser source (2) through a focusing lens (3). The reactor had a set of high-pressure windows, in particular, in the cover of the reactor for visual observation (7), as well as a set of high-pressure ports for introducing gases and various sensors (8).

The body and all parts of the reactor were composed of nonmagnetic S316 stainless steel. As optical ports, quartz glass with a thickness of 10 mm and a diameter of 14.6 mm was used. A palladium metal target (4) was mounted inside the reactor in the fluoroplastic holder; in the lower part of the reactor, there was a recess for the magnetic armature (6) and the solid support for the catalytic nanoparticles (5). The reactor, placed on a magnetic stirrer, was heated by two 200 W clamped heating elements. A pressure sensor, a thermocouple, a high-pressure inlet and outlet valve, and a safety valve were connected to the high-pressure ports. To smoothly lower the pressure over a long time (from 1 to 12 h), a precision valve was used.

### 3.2. Materials

A plate composed of pure palladium (99.96%) with a size of 10 × 10 × 1 mm^3^ and a mass of 1.2 g was used as a target for ablation (4). Carbon dioxide with a purity of 99.99% was used as a medium. The catalyst carrier was powdered γ-alumina with a particle size of 250–500 μm (5). The use of such a sufficiently coarse-grained carrier was previously considered [3,4] and its suitability for these purposes was proven since it has a high chemical and thermal stability (melting point 2050 °C) and a sufficient degree of hardness (according to the Mohs-9 scale). In this case, other important parameters of this material were as follows: The density of γ-Al_2_O_3_ is 3.68 g/cm^3^, the specific surface area (Ssp) is 112 m^2^/g, and the porosity (Vpore) is 0.195 cm^3^/g with an average pore size of about 3.5 nm. This carrier is readily available and can easily be separated into discrete microparticle size fractions. Note that in catalysis of liquid-phase processes, fractions with a particle size in the range of 100–500 microns are mainly used.

### 3.3. Laser Exposition

An Nd:YAG (2) Lotis LS-2138TF laser source (Lotis, Minsk, Belarus) with a Q-switching mode, pulse duration of 15 ns, frequency of 50 Hz, and wavelength and energy of 1064 nm (up to 270 mJ/pulse) and 532 nm (up to 110 mJ/pulse) was used. The radiation was focused using a quartz lens with a focal length of 7 cm. The size of the laser spot on the target was ~0.5 mm.

In this configuration, the energy density in one pulse was up to 140 J/cm^2^ (1064 nm) and up to 60 J/cm^2^ (532 nm); the power density was up to 7 × 10^3^ W/cm^2^ (1064 nm) and 3 × 10^3^ W/cm^2^ (532 nm); the instantaneous power density per pulse was up to 9 × 10^9^ W/cm^2^ (1064 nm) and 4 × 10^9^ W/cm^2^ (532 nm).

The laser exposure on Pd target time was varied within 10–15 min, after which a magnetic stirrer was switched on, causing powdered alumina to fill the internal cavity of the reactor, including the laser radiation region. In this mixed mode, the ablation process continued for another 10–15 min. After the end of the ablation process, the reactor was cooled to room temperature, the pressure was slowly reduced (2–12 h), and Al_2_O_3_ powder was removed for further studies.

### 3.4. Methods of Characterization

To study carrier samples with palladium particles, scanning electron microscopy (SEM), transmission electron microscopy (TEM), high-angle annular dark-field imaging scanning transmission electron microscopy (HAADF STEM), and energy-dispersive X-ray (EDX) spectroscopy using SIRIS (ThermoFisher Scientific, Waltham, MA, USA), FEI Scios (FEI, Waltham, MA, USA), and Phenom PROX (ThermoFisher Scientific, Waltham, MA, USA) devices were used. ImageJ software was used to analyze the obtained images.

The palladium content on the carrier was determined by atomic absorption spectroscopy (AAS) on an AAnalyst 400 instrument (Perkin Elmer, Waltham, MA, USA). The catalytic activity of the obtained composite samples was determined in the liquid-phase hydrogenation reaction of a model substrate, diphenylacetylene (DPA), in an autoclave-type reactor with vigorous stirring of the reaction mixture at 28 °C and a hydrogen pressure of 1 MPa. The reactor had a sampling system and an electronic pressure sensor to control the flow of hydrogen and determine the kinetics of the reaction. It was previously shown that the selected conditions ensure the reaction in the kinetic region [34]. The reaction mixture was analyzed using a Crystal 5000 gas chromatograph (Chromatek, Moscow, Russia).

In kinetic studies, it is generally accepted that the speed of any process is determined by the tangent of the angle of inclination drawn on the initial points of the kinetic curves:W_observ_ = H_2_(_absorbed_)[mol]/t[min](1)
where H_2_(_absorbed_)—the number of absorbed hydrogen, [mol], is equal to the number of the reacted subsurface, [mol].

The specific catalytic activity is expressed as turnover frequency (TOF), which was determined as the ratio of the number of g-atoms [mol] of the reacted substrate to the number of g-palladium atoms [mol] per unit time:TOF [min^−1^] = W_observ_/Me(2)
where Me is the amount of metal, [mol].

The error in determining the quantitative characteristics of the hydrogenation reaction was 5–8%.

## 4. Conclusions

In this study, a new approach to the formation of a nanosized carrier–metal composite based on laser ablation of palladium metal in supercritical carbon dioxide in the presence of an inorganic solid carrier of γ-alumina was proposed and developed. Using electron microscopic methods, the nanoscale characteristics of the obtained materials were determined. Palladium nanocomposites exhibited high activity as heterogeneous catalysts in the process of liquid-phase hydrogenation of an unsaturated hydrocarbon, diphenylacetylene.

## Figures and Tables

**Figure 1 molecules-25-05807-f001:**
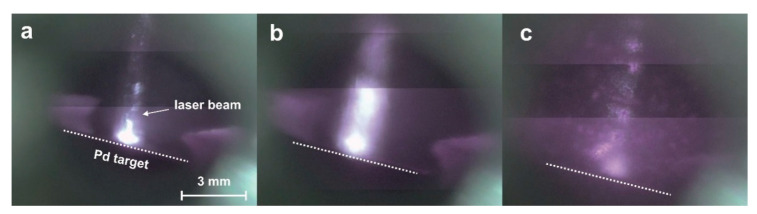
Photographs of a palladium target during laser ablation in supercritical CO_2_ (scCO_2_): (**a**) the beginning of the ablation process, (**b**) after ~5 min of ablation in the first stage (the stirrer is not turned on), and (**c**) the second step after turning on the stirrer. The dashed white line indicates the surface of the target.

**Figure 2 molecules-25-05807-f002:**
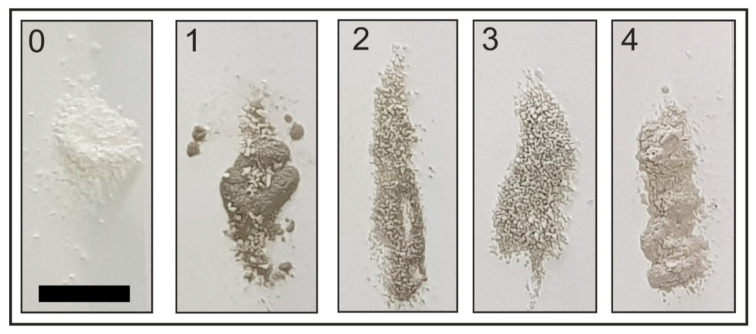
Photos of the samples obtained as a result of laser synthesis. Scale bar = 10 mm. 0 is initial Al_2_O_3_ powder. 1,2,3,4 correspond to the numbers of the samples described in Table 1.

**Figure 3 molecules-25-05807-f003:**
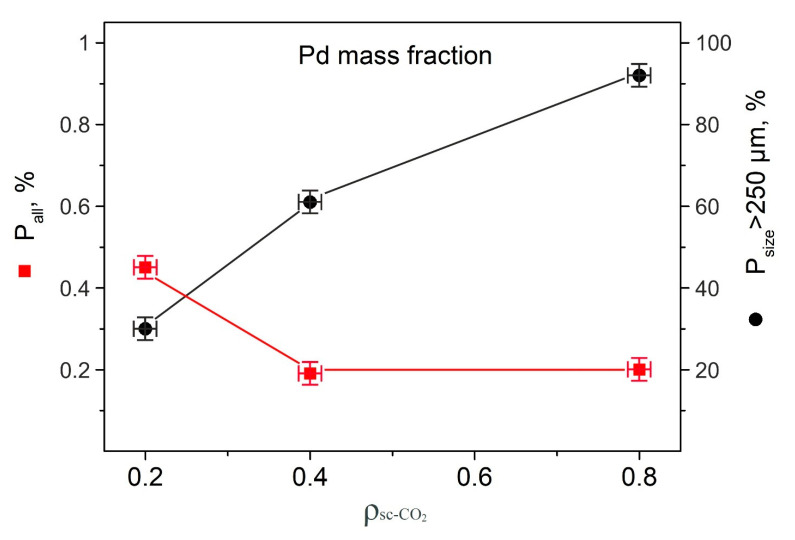
Dependence of the palladium fraction in Al_2_O_3_ particles on the density of the scCO_2_ medium during laser synthesis.

**Figure 4 molecules-25-05807-f004:**
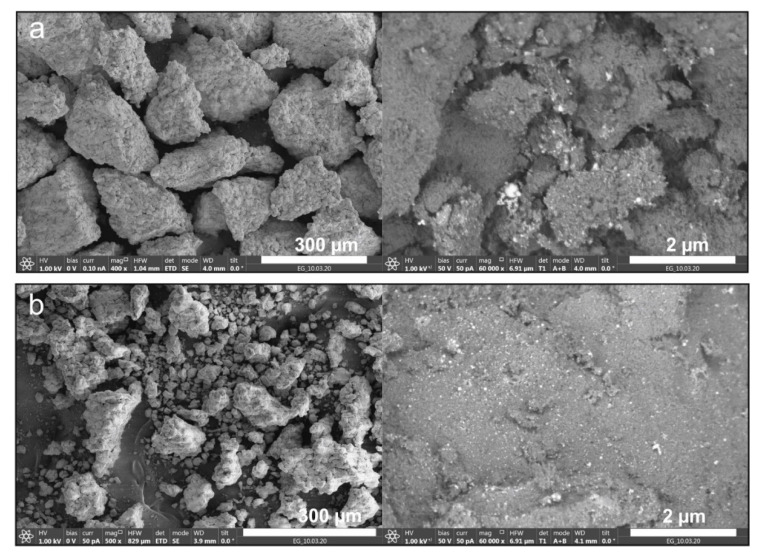
SEM images of Al_2_O_3_ microparticles with deposited Pd particles for (**a**) fraction 0.25–0.5 (sample 5-1) and (**b**) fraction 0.1–0.25 (sample 5-2).

**Figure 5 molecules-25-05807-f005:**
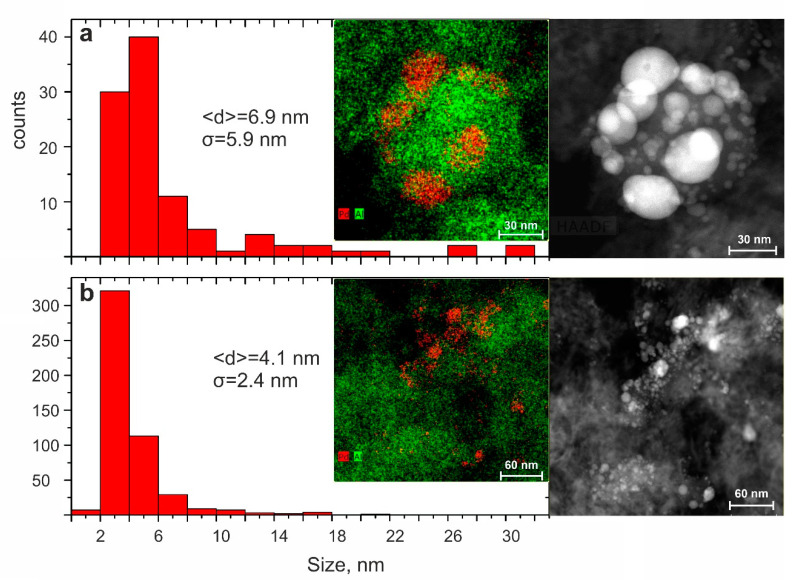
TEM images of fragments of (**a**) fraction 0.25–0.5 (sample 5-1) and (**b**) fraction 0.1–0.25 (sample 5-2), and the corresponding size distribution of palladium nanoparticles.

**Figure 6 molecules-25-05807-f006:**
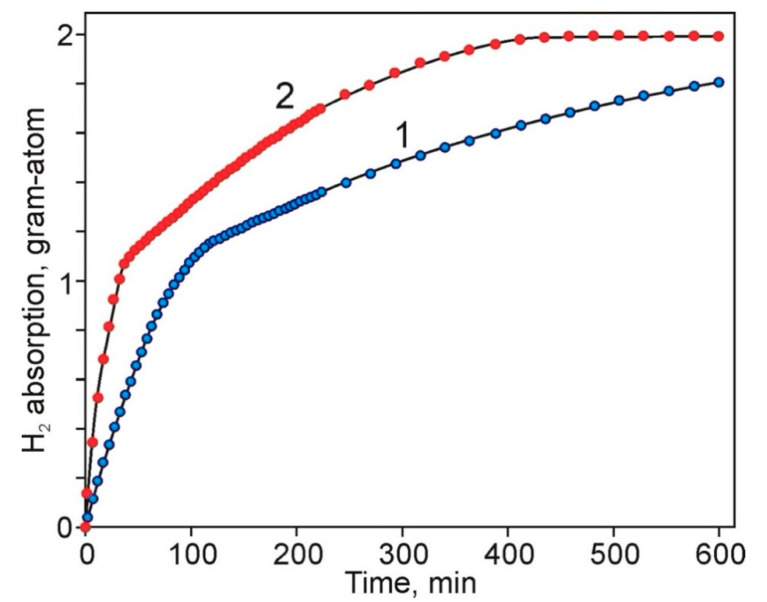
Kinetic curves of hydrogen absorption during hydrogenation of diphenylacetylene (DPA) on Pd/Al_2_O_3_ with the (1) fraction 0.25–0.5 (sample 5-1) and (2) fraction 0.1–0.25 (sample 5-2). Parameters: P (H_2_) = 1 MPa., T = 28 °C, [C≡] = 0.16 mol/l, mcat = 5 mg.

**Figure 7 molecules-25-05807-f007:**
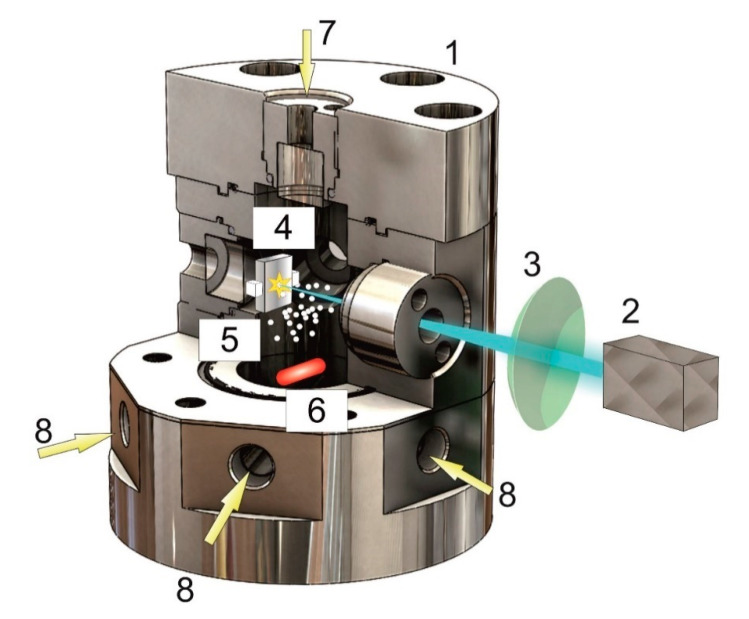
Scheme of the main elements of the experimental installation. 1—high-pressure reactor, 2—laser radiation source, 3—focusing lens, 4—palladium target in a fluoroplastic holder, 5—suspended Al_2_O_3_ particles, 6—magnetic stir bar, 7—a window for visual observation of the process, and 8—high-pressure ports for introducing gas and sensors.

**Table 1 molecules-25-05807-t001:** Pd content on Al_2_O_3_ depending on the medium density and radiation parameters during laser synthesis. T, temperature; P, pressure.

No.	Medium Density, g/cm^3^ (T, °C; P, atm)	Ablation Time, min	Time of Ablation with a Stirrer, min	Wavelength λ, nm (Pulse Energy, mJ)	Pd Content, wt %	Proportion of Pd in Al_2_O_3_ Fraction > 0.25 mm, %
1	0.2 (70; 100)	10	10	1064 (270)	0.45	30
2	0.4 (70; 130)	10	10	1064 (270)	0.19	61
3	0.8 (50; 210)	10	10	1064 (270)	0.20	92
4	0.8 (50; 210)	15	15	532 (110)	0.35	-

**Table 2 molecules-25-05807-t002:** Characterization of the synthesized sample 5 for research by SEM and TEM.

Al_2_O_3_ Fractions, mm	Initial (Sample 5)	Fraction 0.25–0.5 (Sample 5-1)	Fraction 0.1–0.25 (Sample 5-2)	Fraction Less Than 0.1
Mass of fraction, g	1.5	1.0	0.3	0.2
Pd content, wt %	0	0.1	0.47	Not determined

**Table 3 molecules-25-05807-t003:** The activity of various fractions of sample 5 as a catalyst in the hydrogenation of DPA.

Fraction #	Fraction Size, mm	Pd Content, wt %	Wobserv. × 10^5^, mol/min		TOF, s^−1^	
			1st Stage	2nd Stage	1st Stage	2nd Stage
5-1	0.25–0.5	0.1	0.60 ± 0.15	0.08 ± 0.02	2.11 ± 0.2	0.28 ± 0.05
5-2	0.1–0.5	0.47	1.60 ± 0.20	0.15 ± 0.02	1.20 ± 0.15	0.12 ± 0.04

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
