# Peer review of "Synthesis of Supported Heterogeneous Catalysts by Laser Ablation of Metallic Palladium in Supercritical Carbon Dioxide Medium"

_molecules, 2020, doi:10.3390/molecules25245807_

Round 1
Reviewer 1 Report
Molecules - Comments and Suggestions for Authors
The paper presents an interesting study to obtain a supported heterogeneous catalyst using laser ablation technique on metallic palladium and using microparticles of Al2O3 in supercritical CO2.
My suggestion is to consider publishing the paper after major revisions. Here revision comments:
- the motivation of using super critical CO2 should be briefly explained in the introduction.
- In “Materials and Methods” section the Installation part is very well explained, but in the “Result and Discussion” session the description of the obtained samples is a bit confusing (the risk is that the reader is a bit lost).
- The main revision should be done, in my opinion, to data reported in Fig.7. The authors say that “The catalytic activity of the systems was determined using the slope of the kinetic curves - Wobserv, mol/min”. If the data have been fitted by a curve, it is suggested to write the math expression. From data in Fig.7 it seems that fraction 2 (0.47% wt Pd) shows a better response in the H2 absorption, while later the authors say that “despite the almost five-fold excess palladium content in the fraction of the sample of smaller size, its specific catalytic activity was almost two times lower compared to the larger fraction.”, considering the TOF results (I guess). It is really suggested to explain better this crucial point of the paper, for example explaining how the TOF has been obtained.
- Other important revision on Fig.7: you are comparing only 2 set of data. In order to observe a real trend, it would be good to insert at least a third one.
- Minor misspelling or words to check: line 57 “is shows”; line 116 “support for he”.
Author Response
Dear Reviewer,
We are grateful for your careful reading of our article.
Q1. The motivation of using super critical CO2 should be briefly explained in the introduction.
Answer: Thank you for your comment. We have added the necessary explanatory comments.
Added text:
The unique properties of the scCO2 medium, which has a very low viscosity and high molecular mobility, provides effective drift of metal nanoparticles arising during ablation to the surface of carriers and deposition on the surface and in their pores
Q2. In “Materials and Methods” section the Installation part is very well explained, but in the “Result and Discussion” session the description of the obtained samples is a bit confusing (the risk is that the reader is a bit lost).
Answer: Thank you for your comment. In the discussion section, the presentation of the material has been improved, additional clarifications have been added, etc.
Q3. The main revision should be done, in my opinion, to data reported in Fig.7. The authors say that “The catalytic activity of the systems was determined using the slope of the kinetic curves - Wobserv, mol/min”. If the data have been fitted by a curve, it is suggested to write the math expression. From data in Fig.7 it seems that fraction 2 (0.47% wt Pd) shows a better response in the H2 absorption, while later the authors say that “despite the almost five-fold excess palladium content in the fraction of the sample of smaller size, its specific catalytic activity was almost two times lower compared to the larger fraction.”, considering the TOF results (I guess). It is really suggested to explain better this crucial point of the paper, for example explaining how the TOF has been obtained.
Answer:
Thank you for your comment. We have revised the description of our method for evaluating catalytic activity. We provide formulas with specific descriptions in section 2.4. Methods of Characterization.
Added text:
In kinetic studies, it is generally accepted that the speed of any process is determined by the tangent of the angle of inclination drawn on the initial points of the kinetic curves:
Wobserv = H2(absorbed)[mol] / t[min],
where H2(absorbed) - the number of absorbed hydrogen, [mol], is equal to the number of the reacted subsurface, [mol].
The specific catalytic activity is expressed as turnover frequency (TOF), which was determined as the ratio of the number of g-atoms [mol] of the reacted substrate to the number of g-palladium atoms [mol] per unit time:
TOF [min-1] = Wobserv / Me,
where Me is the amount of metal, [mol].
Q4. Other important revision on Fig.7: you are comparing only 2 set of data. In order to observe a real trend, it would be good to insert at least a third one.
Answer: In sample 3 (Table 2), with an Al2O3 particle size less than 0.1 µm, the palladium content was not determined and was not used in hydrogenation catalysis. This fraction had too small particle sizes, which made it unsuitable for catalytic reactions.
Q5. Minor misspelling or words to check: line 57 “is shows”; line 116 “support for he”.
Answer:
Thank you for correcting the text in terms of translation.
The text “is shows” has been changed to “shows”; “support for he” has been changed to “support for the”.
Good luck, and most importantly, good health!
On behalf of authors, Dr. Nikita Minaev
Reviewer 2 Report
In this study, the formation of a small Pd nanoparticles onto the micrometer sized gama-alumina substrate by ns-laser ablation of Pd metal in the supercritical carbon dioxide was presented and characterized by several methods. As shown, these Pd nanocomposites exhibited high activity towards hydrogenation of an unsaturated diphenylacetylene. Although the synthesis approach used herein is well known, this paper is well written and sound. The results presented can be useful for scientists working in the nanomaterials composite fabrication for practical usage. However, to increase the scientific impact of this work, the discussion on the reasons of such composite activity increase should be presented. In addition, the results on the duration of activity are needed for successful application.
Author Response
Dear Reviewer,
We are grateful for your careful reading of our article.
Thanks for the helpful comments. We used them and corrected the text of the article.
- The activity of palladium catalysts in the hydrogenation of unsaturated carbon-carbon bonds has long been well known. The main problem of increasing their efficiency is to increase the specific catalytic activity, i.e. participation in acts of catalysis of all metal atoms, in contrast to traditional, massive heterogeneous contacts, where only surface-active centers take part in catalytic reactions. Another reason for the rather high activity of the proposed catalytic systems can be associated with the method of deposition of palladium particles on the surface or in the internal pores of the support from their colloidal solution in supercritical carbon dioxide. Most likely, it is this method that provides a uniform and uniform distribution of metal particles in the support matrix, which leads to high activity of the composites.
- Of course, the authors agree with the reviewer that in practice the stability of the catalytic action of composites is of great importance. Testing of catalyst sample in successive hydrogenation runs is planned for the near future.
Added text (discussion section):
Palladium catalysts supported on carriers of various natures were used for the hydrogenation of diphenylacetylene in a number of works [33–35]. For this purpose, Markov et al. prepared bimetallic Pd – Cu catalysts by depositing solutions of palladium and copper salts on metal oxides, including Al2O3, followed by drying, calcining, and activating the samples in a stream of hydrogen [33,34]. The specific catalytic activity of these systems in DPA hydrogenation was 0.16-5.77 s-1, depending on the Cu / Pd ratio. In [35], palladium was deposited in a matrix of hyper crosslinked polystyrene, where at the first stage in the medium of supercritical carbon dioxide, palladium compounds were deposited on the support, which in the second stage were reduced to metal with molecular hydrogen. In this case, the specific activity in DPA hydrogenation was 0.57 s-1. The palladium composites obtained in this work are comparable or to exceed the known catalysts in activity, but the process of their preparation includes only one stage, is distinguished by the absence of solvents, and therefore is more environmentally friendly. Also, it allows you to influence the properties of the resulting systems by varying the conditions of laser ablation. It should be noted that the rather high activity of the proposed catalytic systems can be associated with the method of deposition of palladium particles on the surface or in the internal pores of the support from a colloidal solution in supercritical carbon dioxide. Most likely, it is this method that provides a uniform and distribution of metal particles in the support matrix, which leads to high activity of the composites.
The results presented in the article are the first example of obtaining supported heterogeneous catalysts for the hydrogenation reaction by laser ablation of a metal in a supercritical carbon dioxide medium with the deposition of the resulting metal particles on the surface of the carrier.
Good luck, and most importantly, good health!
On behalf of authors, Dr. Nikita Minaev
Reviewer 3 Report
The paper molecules-1014845 deal with the synthesis and, secondly the characterization of laser ablated nanoparticles Pd on Al2O3.
The experimental procedure to obtain the samples is clear and well described. The introduction covers the important aspect of the research and previous paper on similar topic.
There are some remarks that should, anyway, be considered.
It is not clear the distinction between the samples studied :
Table 1 seems to report the experimental conditions to achieve different samples, but only four samples are clearly reported, while the main characterization reported in table 2 and in figure 5, 6 and 7 are related to “sample 5”. The description of this sample is briefly reported at line 248 but it is not clear how its results could be correlated with the other samples.
A clearer discussion that correlates the morphological and physical properties with catalytic performance is required.
Line 246: It is not clear why the different wavelength (532 nm) generates a “more intense destruction of the powders”. I see that a longer duration of treatment is easier to understand. More tests to verify the two hypotheses could be performed, otherwise it is not clear the reason to use the 532 nm in the ablation process.
I suggest including in table 3, the physical differences between the two fractions. It would be easier to follow the discussion.
About the performance as heterogenous catalysts, the characterization is rather low, and the discussion is poor. As already stated, the use of the obtained samples as catalyst should be discussed as a function of the different morphologies and physical parameters. It is not clear why the performances have been tested only on two fractions of a specific sample (and not the other samples or different fractions).
Summarizing, the synthesis, ablation, procedure is very clear and well discussed and with minor review this part would be useful for the readers that are involved in this topic. On the other side, I see that major revision are needed to correlate the samples obtained with the catalytic performances.
Author Response
Dear Reviewer,
We are grateful for your careful reading of our article.
Q1. It is not clear the distinction between the samples studied:
Q2. Table 1 seems to report the experimental conditions to achieve different samples, but only four samples are clearly reported, while the main characterization reported in table 2 and in figure 5, 6 and 7 are related to “sample 5”. The description of this sample is briefly reported at line 248 but it is not clear how its results could be correlated with the other samples.
Answer:
Thank you for your comment. We've adjusted the text for better reading
Table 1 shows the results on the content of palladium in samples obtained by laser ablation under various irradiation conditions: the density of the medium scCO2 and the wavelength of laser irradiation (samples 1 - 4). Sample 5 was obtained under conditions practically equal to those for sample 2 (Table 1); the density of the medium is 0.3 g / cm3 (in sample 2 the density is 0.4 g / cm3), the laser wavelength is the same (1064 nm), the ablation time without and with the stirrer is the same (10 + 10 min). This sample 5 after its fractionation (samples 5-1 and 5-2) was studied by electron microscopy and tested in the hydrogenation reaction. The revised numbers of samples 5-1 and 5-2 are included in the article (Tables 2 and 3, Fig. 5-7).
Added text:
To characterize PdNP alumina deposited on microparticles by electron microscopy and tested in the hydrogenation reaction, sample 5 was prepared by laser ablation of palladium at a moderately low fluid density (0.3 g/cm3, 70 °C, and 120 bar CO2) for 10 min of ablation and 10 min in mixed mode. This sample was divided into fractions (0.25-0.5mm, sample 5-1 and 0.1-0.25mm, sample 5-2) according to the size of the particles in which the mass content of palladium was determined (Table 2).
Q3. A clearer discussion that correlates the morphological and physical properties with catalytic performance is required.
Answer:
The morphological characteristics of catalytically active samples have not yet been studied. As for the physical properties, they are mainly determined by the size characteristics of the metal particles on the surface of the carrier. The corresponding comment is inserted into the article after Figure 7 to Table 3.
Added text:
The high activity of Pd / Al2O3 composites is most likely associated with the size parameters of palladium particles on the support, which, according to TEM data, are 2-8 nm with an almost unimodal particle size distribution.
Q4. Line 246: It is not clear why the different wavelength (532 nm) generates a “more intense destruction of the powders”. I see that a longer duration of treatment is easier to understand. More tests to verify the two hypotheses could be performed, otherwise it is not clear the reason to use the 532 nm in the ablation process.
Answer:
Thank you for your comment. The wavelength of laser irradiation was changed to check how this parameter affects the process of obtaining palladium nanoparticles and the efficiency of their deposition on the support. We have added explanations in the text to explain the results obtained with 532nm laser radiation.
Added text:
However, the size of the large powder fraction after treatment with laser radiation of ? = 532 nm significantly decreased. As a result, this fraction completely lacked large particles with a size> 0.25 mm (Table 1). Therefore, this fraction was not used in further studies of catalytic activity. We believe that the significantly stronger grinding of fraction 4 compared to fractions 1-3 is due to two reasons. The first reason is associated with mechanical grinding: to prepare this fraction, the alumina particles were stirred with a magnetic stirrer for a longer time (15 min instead of 10 min). The second reason is due to laser action: when obtaining fraction 4, particles were acted upon by laser pulses with ? = 532 nm instead of = 1064 nm (Table 1). Shorter wavelength radiation due to the higher absorption coefficient in the alumina material contributes to greater heating and destruction of microparticles [34].
[34]: Xie, X.Z.; Pan, Z.Y.; Wei, X.; Huang, F.M.; Hu, W.; Hong, M.H. Hybrid micromachining of transparent sapphire substrate by pulsed green laser irradiation. J. Laser Micro Nanoeng. 2011, 6, 209–213, doi:10.2961/jlmn.2011.03.0007.
Q5. I suggest including in table 3, the physical differences between the two fractions. It would be easier to follow the discussion.
Answer.
Thank you for your comment. We have adjusted the content of the tables for a more convenient perception of information
Q6. About the performance as heterogenous catalysts, the characterization is rather low, and the discussion is poor. As already stated, the use of the obtained samples as catalyst should be discussed as a function of the different morphologies and physical parameters. It is not clear why the performances have been tested only on two fractions of a specific sample (and not the other samples or different fractions).
Answer.
Thank you for your comment. We have corrected the text of the article. A deeper and more comprehensive study of the physicochemical properties of catalysts, their relationship with activity in various reactions, the stability of their catalytic action in successive hydrogenation experiments is also planned for the future.
Added text (discussion section):
The results presented in the article are the first example of obtaining supported heterogeneous catalysts for the hydrogenation reaction by laser ablation of a metal in a supercritical carbon dioxide medium with the deposition of the resulting metal particles on the surface of the carrier.
Q7. Summarizing, the synthesis, ablation, procedure is very clear and well discussed and with minor review this part would be useful for the readers that are involved in this topic. On the other side, I see that major revision are needed to correlate the samples obtained with the catalytic performances.
Answer.
Thank you for your comment. We have corrected the text of the article.
Added text (discussion section):
Palladium catalysts supported on supports of various natures were used for the hydrogenation of diphenylacetylene in a number of works [33,35,36]. For this purpose, Markov et al. prepared bimetallic Pd – Cu catalysts by depositing solutions of palladium and copper salts on metal oxides, including Al2O3, followed by drying, calcining, and activating the samples in a stream of hydrogen [33,35]. The specific catalytic activity of these systems in DPA hydrogenation was 0.16-5.77 s-1, depending on the Cu / Pd ratio. In [36], palladium was deposited in a matrix of hyper crosslinked polystyrene, where at the first stage in the medium of supercritical carbon dioxide, palladium compounds were deposited on the support, which in the second stage were reduced to metal with molecular hydrogen. In this case, the specific activity in DPA hydrogenation was 0.57 s-1. The palladium composites obtained in this work are comparable to or exceed the known catalysts in activity, but the process of their preparation includes only one stage, is distinguished by the absence of solvents, and therefore is more environmentally friendly. Also, it allows you to influence the properties of the resulting systems by varying the conditions of laser ablation. It should be noted that the rather high activity of the proposed catalytic systems can be associated with the method of deposition of palladium particles on the surface or in the internal pores of the support from a colloidal solution in supercritical carbon dioxide. Most likely, it is this method that provides a uniform and uniform distribution of metal particles in the support matrix, which leads to high activity of the composites.
[35]: P.V.Markov, G.O.Bragina, G.N.Baeva, A.V.Rassolov, I.S.Mashkovsky and A.Yu.Stakheev. Highly Selective Pd–Cu/α-Al2O3 Catalysts for Liquid-Phase Hydrogenation: The Influence of the Pd : Cu Ratio on the Structure and Catalytic Characteristics. Kinet. Catal. 2018. V. 59. N 5. P. 601-609. DOI: 10.1134/S0023158418050105
Good luck, and most importantly, good health!
On behalf of authors, Dr. Nikita Minaev
Round 2
Reviewer 1 Report
The authors addressed all the revision requests and the paper can be published in this version.
Reviewer 3 Report
The authors answered adequately to the previous concerns.
In my opinion the paper is now suitable to be published.